# Human Placenta Exosomes: Biogenesis, Isolation, Composition, and Prospects for Use in Diagnostics

**DOI:** 10.3390/ijms22042158

**Published:** 2021-02-22

**Authors:** Evgeniya E. Burkova, Sergey E. Sedykh, Georgy A. Nevinsky

**Affiliations:** 1SB RAS Institute of Chemical Biology and Fundamental Medicine, 630090 Novosibirsk, Russia; sedyh@niboch.nsc.ru (S.E.S.); nevinsky@niboch.nsc.ru (G.A.N.); 2Department of Natural Sciences, Novosibirsk State University, 630090 Novosibirsk, Russia

**Keywords:** placenta, trophoblast, exosomes, placental exosomes, isolation, protein, miRNA, lipid, biomarkers

## Abstract

Exosomes are 40–100 nm nanovesicles participating in intercellular communication and transferring various bioactive proteins, mRNAs, miRNAs, and lipids. During pregnancy, the placenta releases exosomes into the maternal circulation. Placental exosomes are detected in the maternal blood even in the first trimester of pregnancy and their numbers increase significantly by the end of pregnancy. Exosomes are necessary for the normal functioning of the placenta and fetal development. Effects of exosomes on target cells depend not only on their concentration but also on their intrinsic components. The biochemical composition of the placental exosomes may cause various complications of pregnancy. Some studies relate the changes in the composition of nanovesicles to placental dysfunction. Isolation of placental exosomes from the blood of pregnant women and the study of protein, lipid, and nucleic composition can lead to the development of methods for early diagnosis of pregnancy pathologies. This review describes the biogenesis of exosomes, methods of their isolation, analyzes their biochemical composition, and considers the prospects for using exosomes to diagnose pregnancy pathologies.

## 1. Introduction

Exosomes are vesicles of 40–100 nm in size released from cells after fusion of multivesicular bodies (MVBs) with the plasma membrane. Exosomes have a cup shape when studied under an electron microscope, with a floating density ranging from 1.13 to 1.19 g/mL. Due to the endosomal origin, exosomes are enriched with endosomal membrane markers CD63, CD9, and CD81 [1]. These vesicles are often enriched in cholesterol, sphingomyelin, glycosphingolipids, and phosphatidylserine [2]. Complex polysaccharides carry the terminal residues of galactose, N-acetylglucosamine, mannose, α-D-mannosyl, and α-D-glucosyl groups located on the surface of exosomes [3,4]. Exosomes have been described in various biological fluids: blood, urine, milk, tears, saliva, ascetic, synovial, bronchoalveolar, amniotic, semen, and vaginal fluids [1,5,6,7,8,9]. These nanovesicles were found to regulate the immune response, antigen presentation, and regulation of vascular homeostasis [1].

It has been shown that during implantation, embryos secrete exosomes and other extracellular vesicles [10,11]. These exosomes can interact with the mother’s [10] and embryonic [11] cells.

At the sixth week of pregnancy, the placenta begins to form [12]. The placenta contains trophoblast cells, which are differentiated in villous and extravillous trophoblasts [12]. Trophoblasts exhibit protection, nutrition, and respiration of the fetus and the production of hormones. Throughout pregnancy, the trophoblast secretes exosomes into the maternal circulation [13,14]. Placental alkaline phosphatase (PLAP) is a specific marker of syncytiotrophoblast exosomes [15]. PLAP-containing exosomes are described in the first trimester of pregnancy, and their number increases significantly toward the end of pregnancy [14]. Placental exosomes transport various proteins, miRNAs, and lipids over a considerable distance from the place of origin. Thus, they can change the activity of neighboring cells or act remotely by transferring cargo through biological fluids. The study of human placenta exosomes, especially trophoblast-derived exosomes, has attracted considerable interest in recent years due to their presumed important role during pregnancy. Placental exosomes can play an essential role in preventing fetal rejection [16,17] and complicating pregnancy [18,19].

Recently, there has been a growing number of women of reproductive age with various pregnancy complications, such as preeclampsia (PE) and gestational diabetes [20]. The expected consequences of these pathologies are stillbirth, fetal growth restriction, and premature birth. At present, the structure, the composition, and the biological activity of placental exosomes, and the features of their circulation in health [17,21,22] and pathologies [23,24] are being studied. Analyzing placental exosomes circulating in a pregnant woman’s blood and identifying their contents will allow one to develop new tools for non-invasive diagnostics of the placenta’s functional and structural state. This review focuses on the biogenesis of placental exosomes, their isolation and characterization, the analysis of their biochemical composition, and prospects for their use in the diagnosis of pregnancy pathologies.

## 2. Biogenesis and Secretion of Exosomes

### 2.1. Biogenesis of Exosomes

Exosomes are formed by the endosome system involved in sorting and transporting cellular vesicles to their location (Figure 1). Early endosomes mature into late endosomes. During this process, the endosome contents are acidified due to the V-ATPase proton pump’s operation. They also lose Rab5 and acquire Rab7 and Rab9, which, together with the mannose-6-phosphate receptor, are markers of late endosomes [25]. Intraluminal vesicles (ILVs) are constitutively formed by inward budding of the endosome membrane and are accumulated in endosome cavities [26].

The exosome biogenesis in human syncytiotrophoblasts has been studied mainly at the ultrastructural level using MICA/B, ULBP 1–5, FasL, TRAIL, PD-L1, also the specific exosomal markers [27,28]. These molecules are absent on the plasma membrane of the syncytiotrophoblast. However, FasL, TRAIL, PD-L1, MICA/B, and ULBP 1–5 were expressed on the limiting membrane of multivesicular bodies (MVBs) and the membrane of intraluminal nanosized vesicles in MVBs [15,27,28]. These molecules may be sorted from the Golgi to MVBs [15]. MICA and MICB were expressed on the apical membrane of syncytiotrophoblast and exosomes inside MVBs [28].

There are two mechanisms of formation of ILVs: ESCRT-dependent and ESCRT-independent [1].

#### 2.1.1. ESCRT-Dependent Mechanism of Exosomes Formation

The best-described mechanism for forming MVBs and ILVs occurs with the sorting complex of endosomes required for transport—the ESCRT complex. The ESCRT complex consists of four molecular complexes (ESCRT-0, -I, -II, and -III) [29,30]. The ESCRT complex members were shown localized to MVBs of the syncytiotrophoblast [15]. The assembly of the ESCRT complex begins with the binding of ESCRT-0 to phosphatidylinositol-3-phosphate. The ESCRT-0 complex recognizes and binds ubiquitinylated transmembrane proteins of the endosome and is also responsible for loading them into ILVs. Sorting of FasL, TRAIL MICB, ULBP4, and 5 may be controlled by their ubiquitinylation in MVBs of syncytiotrophoblast [15,27]. GPI-linked ULBP 1–3 molecules and the transmembrane MICA are preferably expressed in lipid rafts at the cell surface and may be recycled and sorted into syncytiotrophoblast MVBs [15]. The ESCRT-0 complex contains the protein hepatocyte growth factor (HRS), which recognizes monoubiquitinylated proteins and binds to STAM (signal transduction adaptor molecule, the second component of ESCRT-0), Eps15, and clathrin (two non-ESCRT complex proteins). HRS binds to the TSG101 protein of the ESCRT-I complex. Then, the ESCRT-I complex participates in the activation of the ESCRT-II complex, interacting via the adapter protein Alix. ESCRT-I and -II complexes induce vesicle formation by membrane invagination of endosomes. ESCRT-II, together with enzymes that remove ubiquitin, promotes the movement of labeled proteins into ILVs. ESCRT-III is responsible for the vesicle detachment from the membrane into MVBs [29]. The dissociation of the ESCRT complex requires interaction with the AAA-ATPase VPS4B.

#### 2.1.2. ESCRT-Independent Mechanism of Exosomes Formation

Inhibition of neutral sphingomyelinase leading to reduced ceramide biogenesis reduces exosome secretion. Ceramides are assumed to induce plasma membrane invagination inside the MVBs with the formation of ILVs [31]. Phospholipase D2 is also required for exosome biogenesis [32]. It is suggested that the phosphatidic acid formed in the inner layer of the MVBs membrane induces the invagination of the endosome membrane and, thus, the formation of ILVs. 

When ILVs are formed, their contents are loaded, with some proteins and RNAs entering exosomes by passive capture of the cytoplasm contents [26]. However, due to exosomes containing sets of specific proteins and RNAs differing from those of parent cells, these specific proteins and RNAs are believed to be transported selectively to exosomes [30].

### 2.2. Secretion and Interaction of Exosomes with Target Cells

MVBs can either be directed to lysosomes where their content is degraded or transported to the plasma membrane for exosome release. The Rab GTPases control various intracellular vesicular transport stages, such as vesicle formation, movement of vesicles and organelles along the cytoskeleton filaments, and fusion with the membrane [33]. Rab11 is necessary for the secretion of exosomes induced by Ca^2+^-ions [34]. Knockdown of the Rab5a, Rab9a, Rab2b, Rab27a, and Rab2b genes leads to a significant decrease in exosome secretion [35]. It has been shown that Rab7 is required for the secretion of exosomes containing syntenin and ALIX [36]. Placental exosomes contain all these proteins [15]. Syntenin and ALIX are assumed to be involved in the docking of MVBs and the cell plasma membrane, and are necessary for the final fusion of the two membranes, making secretion possible.

After docking of two different intracellular compartments, the SNARE complex proteins SNAP-23, VAMP-7, and VAMP-8 participate in Ca^2+^-regulated fusion of MVBs with the plasma membrane in various cells [37,38]. 

After being released into the extracellular space, exosomes circulate in body fluids until they interact with target cells. Trophoblast exosomes contain fibronectin, a ligand for the α5β1 integrin of macrophages, syncytin-1, and syncytin-2, which bind to the ASCT1, ASCT2, and MFSD2a receptors on trophoblast cells and thus participate in the uptake of exosomes by cells [39]. Placental exosomes express FasL and TRAIL on their surface and trigger apoptosis in Jurkat T cells by suppressing NF-κB, CD3ζ, and Janus kinase 3 expressions [16,21].

The mechanisms of exosome penetration into recipient cells have not been sufficiently studied. However, it has been shown that, depending on the recipient cell type, exosomes enter target cells by fusion with the plasma membrane, macropinocytosis, phagocytosis, and clathrin-dependent endocytosis [40].

## 3. Methods of Exosome Isolation

Many methods are used to isolate exosomes from the placenta and various biological fluids, and new isolation protocols appear in the literature every year. The International Society for Extracellular Vesicles advises detecting specific markers of exosomes, such as CD81, CD9, and CD63 tetraspanins by immunoelectron microscopy, flow cytometry, or Western blot to confirm the exosomal nature of the isolated vesicles [41,42,43]. However, due to detecting all particles in the solution with CD81, CD9, and CD63, Western blot and flow cytometry are not selective enough to analyze membrane and non-membrane structures. Immunoelectron microscopy allows detecting exosomal markers directly on the nanovesicle surface [44]. Transmission electron microscopy can be used to analyze exosome morphology and evaluate the preparation purity [5,44]. TEM (transmission electron microscopy) showing co-isolating impurities in exosome preparations (proteins, protein complexes, microvesicles) can lead to misinterpretation of biochemical contents of insufficiently purified preparations [44]. For example, according to transmission electron microscopy analysis, exosome preparations isolated from blood and various fluids by sequential centrifugations and ultracentrifugation contain “non-vesicles” that correspond in morphology to intermediate- and low-density lipoproteins (20–40 nm) and very-low-density lipoproteins (30–80 nm) [44]. High-density (8–11 nm) and low-density lipoproteins can also be combined with plasma exosomes [45,46]. Lipoproteins of different densities, as well as exosomes, can carry RNA and proteins [47]. Therefore, studying the composition and functions of exosomes contaminated with other structures can lead to false-positive results.

Pregnancy-associated exosomes have been isolated from the blood of pregnant women at different gestational ages [14,23,48,49], ex vivo cultures of placental explants of different gestational ages [18,27], placental perfusate [50] and placental homogenate obtained after term delivery [51,52,53], and trophoblast cells [16,54].

The standard protocol for exosome isolation from biological fluids includes sequential centrifugations at low speeds to remove cellular debris, large vesicles, and ultracentrifugation at 100,000× *g* [55,56]. This protocol was used to purify pregnancy-associated exosomes from different sources. Currently, several variants of this method are used for isolation exosomes from the blood of pregnant women or from the trophoblast cells, for example, ultracentrifugation at 110,000× *g* [16] or 200,000× *g* [13,14]. However, during ultracentrifugation, various types of extracellular vesicles are enriched with protein and protein aggregates. For additional purification of pregnancy-associated exosomes from other vesicles and co-isolating proteins, a density gradient ultracentrifugation [54,55] or a sucrose cushion [18] was used. Most of the exosomes used so far in various studies described above and below were obtained using exactly these methods yielding not pure but exosome-enriched preparations.

Commercial kits have become available in the last several years. For example, ExoQuick (System Biosciences, Palo Alto, CA, USA) is used for isolating exosomes from plasma of pregnant women [19,57,58]. However, in addition to possible non-exosomal structures, the exosome samples contain polymer molecules that are incompatible with mass spectrometry use. Using the ExoQuick kit (System Biosciences, Palo Alto, CA, USA) was shown to produce the most contaminated plasma exosome preparations [59]. Therefore, sucrose density gradient ultracentrifugation for separating placental exosomes was used after ExoQuick (System Biosciences, CA, USA) purification [57].

A method for isolating trophoblast exosomes using a microfluidic chip and ultrasonic waves has been proposed for placentas obtained after delivery [60]. The use of sound waves permits obtaining intact exosomes. However, this method is not specific to exosomes since other vesicles comparable in size to exosomes can be co-isolated.

A chromatographic/immunosorbent procedure was used to specifically isolate syncytiotrophoblast exosomes (PLAP^+^-exosomes) from the maternal peripheral circulation [21]. Total plasma exosomes from the blood of pregnant women were isolated by size-exclusion chromatography, then PLAP^+^-exosomes were separated using binding to specific antibodies coupled to microbeads.

For additional separation of placental exosomes from a large number of various impurities, size-exclusion exclusion chromatography was successfully used in combination with ultracentrifugation and ultrafiltration for a wide range of biological fluids [51]. The main advantage of this method is the high efficiency of removing impurities, such as non-specifically interacting proteins or protein aggregates. Low pressure and gravity-dependent flows allow intact vesicles to be obtained [61].

We have developed a protocol for isolating highly purified exosome preparations from the human placenta, including ultracentrifugation and ultrafiltration through a 0.1 µm filter, supplemented by size-exclusion chromatography on Sepharose 4B [51]. Exosomes obtained according to a standard protocol without gel filtration contain exosomes and many vesicles larger than 100 nm, “non-vesicles”, i.e., rounded particles without a limiting membrane (Appendix A). Size-exclusion chromatography using Sepharose 4B significantly reduces the number of co-isolating proteins and “non-vesicles” (Figure 2) [51]. After size-exclusion chromatography, the preparations of placental exosomes were shown by flow cytometry to contain ~78% of CD9^+^-exosomes and ~74% of CD81^+^-exosomes [51]. The exosome yield after size-exclusion chromatography was comparatively high (up to ~93%). Hence, this method does not lead to the loss of a large part of exosomes.

All major proteins corresponding to exosome peaks after size-exclusion chromatography were identified using MS and MS/MS data of MALDI-TOF mass spectrometric analysis of tryptic hydrolysates after standard SDS-PAGE and 2D-electrophoresis [51,52]. 2D-electrophoresis identified the proteins of the first peak obtained by size-exclusion chromatography of placental exosomes. Interestingly, 28 spots corresponded to only 12 different proteins and their isoforms [51].

Similar results were obtained for horse milk exosomes, purified using differential centrifugations and size-exclusion chromatography [62]. Such extra purified exosome preparations contained only nine major proteins. Still, many impurity proteins co-isolated with exosomes during ultracentrifugation were found in the second and third peaks after size-exclusion chromatography [62]. Thus, gel filtration is an advantageous approach for the additional effective purification of exosomes from impurities, but there is still room for improvement [63].

We used affinity chromatography on anti-CD81-Sepharose of exosome preparations after ultracentrifugation, ultrafiltration, and gel filtration (Appendix A) [52]. The main part of the exosomes was eluted with 0.15 M NaCl, with the resulting exosome preparations containing almost no impurities. Thus, a combination of ultracentrifugation, ultrafiltration, size-exclusion chromatography, and affinity chromatography makes it possible to obtain sufficiently homogeneous exosome preparations, which is very important for subsequently analyzing their biochemical composition and studying their biological activity.

According to most of the purification methods mentioned above, it should be emphasized that only exosome-enriched samples can be prepared unless single-vesicle isolation approaches to be developed are to be used. Different biological fluids contain various high molecular weight aggregates of multiple proteins co-isolated with exosomes under different centrifugations. Human placenta was shown to contain an extremely stable multiprotein complex (~1000 kDa), with sizes comparable to exosomes [64]. This complex possesses nine different catalytic activities: DNase, RNase, ATPase, phosphatase, protease, amylase, catalase, peroxidase (H_2_O_2_-dependent), and oxidoreductase (H_2_O_2_-independent).

Some proteins may interact specifically or non-specifically directly with exosome surface or with receptors embedded in exosome membranes. It is assumed that published works may have significantly overestimated the relative number of proteins and RNA directly inside exosomes [65]. Therefore, it is worth mentioning the critical review of Sverdlov E.D. [66], stating that, in regard to exosomes, an incorrect overestimated quantitative assessment of internal components has been formed.

## 4. Composition of Human Placental Exosomes

Preparations of exosomes isolated from the blood of pregnant women [14,19,23,48], ex vivo cultures of placental explants of the first trimester of normal pregnancy or with pregnant pathology [18,27,28], perfusate [50] or placental homogenate [51,52,53], trophoblast cell cultures [18,54] were used to study the composition and biological role of placental exosomes. In some studies, JEG-3 [18], HTR-8/Svneo [18], and SW71 [54] cell lines were used. JEG-3 is choriocarcinoma, HTR-8/Svneo (from 6–12 weeks of the first-trimester placenta) is SW71 (from 7 weeks of the first-trimester placenta) immortalized trophoblast cells by SV40. These cell lines are used as a model of invasive extravillous trophoblast. In most published studies, placental exosomes are isolated by centrifugations at different accelerations in a density gradient. Thus, their composition and functional activities may be analyzed on preparations containing other non-exosomal vesicles and co-isolated proteins. We will review the literature data on the composition of human placental exosomes and exosomes derived from the trophoblast. 

### 4.1. Proteins of Placental Exosomes

Unfortunately, there are currently only a few proteomic studies on placental exosomes. Depending on the exosome sources and protein analysis methods, several hundred and even thousands of proteins were detected in exosome preparations from the blood of pregnant women, trophoblast cells, or placenta: 140 [18], 160 [67], 200 [68], 282 [54], 340 [13], 1476 [56], 1684 [69]. The most represented exosomal proteins, according to the literature data, are summarized in Table 1.

Regardless of cellular origin, exosomes are believed to contain proteins involved in the formation of MVBs (annexins, Rab proteins of GTPases family, proteins of the ESCRT complex, TSG101, Alix) [15], and also contain tetraspanins (CD9, CD81, CD63) on their surface. Placental exosomes contain placental alkaline phosphatase (PLAP), a specific marker of exosomes released by syncytiotrophoblast [15]. Unlike other exosomes, such as those of immune cells, exosome explants from early normal placentas (8–14 weeks) do not contain the MHC molecule. However, MHC-related molecules MICA⁄B and RAET1⁄ULBP1-5, ligands that activate the NK-cell receptor NKG2D, are expressed on their surface [27]. Exosomes derived from chorionic villi of early (8–14 weeks) and term placentas contain proapoptotic molecules FasL1–4 and TRAIL [13].

When analyzing the data described above and below, account should be taken of the impossibility of obtaining pure exosome preparations using various centrifugations, including gradient ultracentrifugation. As shown above, such preparations may contain a wide variety of impurities, including high-molecular-weight protein complexes.

Partially purified exosomes isolated from first-trimester extravillous trophoblast cells by differential centrifugations and a sucrose density gradient ultracentrifugation contain proteins that can participate in a wide range of cellular processes: heat shock proteins, secreted proteins, cytoskeleton-related proteins, adhesion and fusion proteins, enzymes involved in amino acid metabolism, lipid metabolism, redox reactions, and synthesis of proteins [54]. Furthermore, placenta-specific proteins, such as pregnancy zone protein and chorionic gonadotropin, have been identified [55].

Trophoblast-derived exosomes from first-trimester (8–12 weeks) placentas isolated using differential centrifugations and iodixanol density gradient centrifugation contain 1476 proteins [56]: histones and enzymes involved in DNA replication, mRNA splicing, transcription and translation, and placenta-specific proteins. Interestingly, 63 proteins out of 1684 proteins are exclusively expressed in trophoblastic exosomes from term placentas, such as CD276, syncytin-1, and syncytin-2 [69].

Comparison of the exosome proteome from JEG-3 and HTR-8/Svneo cell lines isolated by a combination of ultracentrifugation and a sucrose density gradient ultracentrifugation showed only 26 proteins out of the identified 84–85 proteins identified to be represented in the exosomes of both cell lines: α-2-macroglobulin, α-2-HS-glycoprotein, β-actin, serum albumin, apolipoproteins E and M, complement C3, coagulation factor V, β-fibrinogen, thrombospondin, filamin A, heat shock proteins HSP90 and HSP70, VEPH1, lactate dehydrogenase, enolase-1, eukaryotic elongation factor-2, phosphoglycerate kinase, moesin, pyruvate kinase, pregnancy zone protein, thrombospondin, β- and ε-activator of tyrosine-3-monooxygenase [18].

It should be noted that mitochondrial, nuclear, Golgi, and reticulum proteins are not transported into endosomes. Therefore, they cannot be part of exosomes [70]. In any case, since the exosomes originate from an independent cellular compartment, their protein composition is not accidental.

The protocol that we developed for obtaining highly purified preparations of human placenta exosomes allowed us to establish that highly purified preparations of placental exosomes contain no more than 13 different reliably detectable major proteins: α- and β-hemoglobin subunits, ferritin, CD-9, CD81, CD63, annexins A1, A2 and A5, cytoplasmic actin-1, serotransferrin, interleukin-1 receptor, α-actinin-4, and PLAP [51]. As noted above, the possibility of exosomes containing serum albumin and IgG antibodies remains a big question [52]. Moreover, many proteins are found in partially purified preparations using different centrifugations [18,54,56,69], which we detected in the second and third peaks of impurities separated from exosomes by size-exclusion chromatography (Appendix A).

Large proteins (>10–12 kDa) of exosomes are usually analyzed by MALDI mass spectrometry of tryptic protein hydrolysates after separating them by 2D-electrophoresis. Small proteins (<10–12 kDa) and peptides cannot usually be detected after SDS-PAGE or 2D-electrophoresis since they either leave the gels during the separation or are washed during Coomassie gel staining. However, various peptides possess many different essential functions. They can be antioxidants, regulators of growth, neuro-mediators, antibiotics, protectors, regulators of blood pressure, levels of calcium, glucose, etc. Exosomes can transfer information to other cells using relatively small proteins and peptides. Recently, we have shown that placental exosomes isolated by a combination of ultracentrifugation, size-exclusion chromatography, and affinity chromatography contain about 27 different small proteins and peptides of 2–12 kDa in addition to the large proteins (>10–12 kDa) [52].

A meta-analysis of six databases containing information on more than 700 proteins found in placental exosomes showed that only three common proteins were found in all databases: albumin, fibronectin-1, and plasminogen activator inhibitor [71]. Different sources can explain conflicting data on protein composition used to isolate placental exosomes, different pregnancy stages, and other exosome purification methods. Therefore, it is necessary to standardize protocols for isolating placental exosomes, including additional steps for their effective purification. The purity of exosome preparations after their purification can be analyzed by transmission electron microscopy. Unfortunately, the question as to which proteins found by the authors of various publications are directly part of the placental exosomes and which of them are co-impurities remains to be answered.

### 4.2. Nucleic Acids of Placental Exosomes

Placental miRNAs are selectively packed into exosomes and microvesicles, secreted into the maternal circulation [72,73]. Thus, placental exosomes transfer genetic information to target cells and regulate their metabolism.

When considering the following data on the analysis of various RNAs, one should also take into account that practically in all various studies, not pure but only exosome-enriched preparations obtained by several types of centrifugations and ultracentrifugation were used. Sverdlov E. D. analyzed all the literature data on the analysis of many thousands of different RNAs found in exosomes and concluded that their number was overestimated. He figuratively stated that these data “would certainly make Amedeo Avogadro cry” [66]. Given below are the data without considering possible errors associated with the contamination of exosome preparations with microvesicles and protein complexes.

Exosomes isolated from trophoblast cells from normal term placentas conditioned the environment to have a miRNA profile similar to the parental cells [72,74]. Trophoblast-derived exosomes from normal term placentas have a high level of expression of placental-specific miRNAs [72]. Many placenta-specific miRNAs are encoded by the chromosome 19 miRNA gene cluster (C19MC). Placenta-specific miRNAs of the C19MC cluster include the following families: miR-512, miR-1323, miR-498, miR-520, miR-515, miR-519, miR-1283, miR-526, miR-525, miR-523, miR-524, miR-518, miR-517, miR-516 [72,75].

Analysis of miRNAs of exosomes isolated from the conditioned medium of chorionic villi of term placentas showed the presence of 456 different miRNAs [74], with the main part of miRNAs being placenta-specific and belonging to the C19MC and C14MC gene clusters and miR-23, miR-127, miR-134, miR-371, miR-372, and miR-373 families also represented. miRNAs of the let-7 family, miR-18a, miR-93, miR-101, miR-141, miR-148, and others expressed not only in the placenta but also in other organs were shown to be also present in exosomes villous trophoblast [74].

Exosomes isolated from plasma and serum of pregnant women using differential centrifugations and ultrafiltration were found to contain DNA of maternal and fetal origin [76]. However, DNA in exosomes is still a controversial issue since DNA can be located inside vesicles of non-exosomal origin, co-isolated with exosomes. DNA can be associated with the surface of exosomes. For example, preparations of exosomes from tear fluid contain DNA. However, treatment of exosomes with DNase I resulted in DNA destruction, so DNA is likely to be localized on the exosome surface [5].

Exosome preparations contaminated with other vesicles and co-isolating proteins were analyzed in many papers. Therefore, some miRNAs can be co-isolated with such proteins and protein complexes. Further, miRNA studies of placental exosomes can significantly expand the prospects for their practical use. 

### 4.3. Lipids of Placental Exosomes

A phospholipid bilayer characteristically envelops exosomes. In placental exosomes, 179 individual phospholipids were detected in eleven major classes: phosphatidylcholine, phosphatidylethanolamine, phosphatidylserine, sphingomyelin, phosphatidylglycerol, phosphatidylinositol, phosphatidic acid, lysophosphatidic acid, cardiolipin, lysophosphatidylcholine, and lysophosphatidylethanolamine [69]. Exosomes contain 2–3 times more cholesterol, sphingomyelin, glycosphingolipids, and phosphatidylserine than their parent cells [77]. Compared to the parent cell, exosomes contain less phosphatidylcholine.

Placental exosomes contain stearic, linoleic, and arachidonic acids, prostaglandins E2, and 15d-PGJ2, which are ten times more abundant in exosomes than in the parent cells [76,78]. Prostaglandin E2 is immunosuppressive; prostaglandin 15d-PGJ2 is a ligand for the PPAPγ nuclear receptor, which plays an essential role in childbirth [79].

Lysophosphatidic acid is present in the membrane of ILVs of MVBs [80] and can also be found in exosomes in small amounts [81]. However, it is assumed that lysophosphatidic acid is not transported to exosomes but is included in the ILVs of MVBs, which subsequently fuse with the lysosome.

Given that some exosome impurities, such as lipoproteins, contain many lipids [82], the data on the lipid composition of nanovesicles can also be considered somewhat speculative.

## 5. Role of Placenta Exosomes in Human Pregnancy

The role of placental exosomes in pregnancy is not fully understood. Many studies may use exosome preparations contaminated with proteins, nucleic acids, and other biological fluid components. Some activities of placental exosomes may be due to impurities. Various possible biological functions of placental exosomes have been described in the literature. When analyzing these data, one should take into account that some exosome functions that were discovered may not be related directly to exosomes but to different protein impurities, DNAs, RNAs, and lipids in exosome preparations.

Oxygen content and glucose concentration play an essential role in regulating biogenesis and secretion of placental exosomes [67,83,84]. The release of exosomes from trophoblast cells is increased under low oxygen tension and high glucose concentration. Exosomes from the first-trimester primary culture of cytotrophoblast cells incubated under hypoxic conditions (pO2 ~6.75 mmHg) most actively stimulate invasion and proliferation of extravillous trophoblast cells [67].

The concentration of exosomes in blood plasma was more than 50-fold greater in pregnant than in non-pregnant women [21]. The concentration of PLAP^+^-derived exosomes in maternal plasma increased with gestational age: first trimester 99.8 ± 5.3 pg/mL; second trimester 397 ± 23; and third trimester 731 ± 35 pg/mL [14]. The concentration of PLAP^+^-exosomes in maternal plasma during the first trimester increased from 6 weeks 70.6 ± 5.7 pg/mL to 12 weeks 117.5 ± 13.4 pg/mL [13].

The total number of exosomes in maternal circulation correlated with the BMI of pregnant women, with 12–25% of circulating exosomes in maternal blood being of placental origin across gestation, and placental exosome contribution to the total exosomes decreases with higher maternal BMI during pregnancy [21]. Exosomes from maternal blood increase TNF-α, IL-6, and IL-8 release from endothelial cells. The effect was higher when exosomes were isolated from obese pregnant women compared to lean and overweight ones.

Trophoblast exosomes may play an essential role in intercellular communications that promote placentation [39,85]. Trophoblast-derived exosomes from term placentas and serum pregnant women contain syncytin-1 and syncytin-2, which bind to the ASCT1, ASCT2, and MFSD2a receptors on trophoblast cells. Therefore, exosomes can be uptaken by these cells, leading to trophoblast cells to fuse and form syncytiotrophoblast [39,85].

Exosomes from the blood of pregnant women with term labor and those with preterm labor have been characterized [21,48]. Incubation of T cells with PLAP^+^-exosomes from the blood of pregnant women resulted in the suppression of CD3-ζ and Janus kinase 3 expressions and activation of caspase 3, which led to weakened of T-cell-mediated responses, confirming the hypotheses of the Th2 cell immune response rejection of the of during pregnancy [21]. Correlation of the CD3-ζ-chain suppression with FasL and programmed death ligand PD-L1 expressions on exosomes may explain the apoptosis induction in target cells [21]. Exosomes from the blood of pregnant women with urgent physiological delivery were found to inhibit IL-2 production by activated T cells compared to exosomes from pregnant women’s blood with term delivery [48]. Furthermore, it was found that exosomes from placental explant cultures of the first trimester (8–16 weeks) transfer NKG2D receptor ligands (MICA⁄B and ULBP1-5) and can suppress the NKG2D receptor on NK, CD8^+^, and γδT cells [27]. Cell internalization of NKG2D receptor induced by exosomes leads to a decrease in receptor-mediated cytotoxicity.

Thus, placental exosomes may be critical for preventing an increased immune response during pregnancy by reducing T cells signaling and cytotoxicity due to suppressing the NKG2D receptor that activates natural killer (NK) cells and providing apoptotic activity via pathways mediated by FasL, TRAIL, and PD-L1 ligands.

Trophoblast exosomes carry immunomodulatory molecules B7-H1 (CD274), B7-H7 (CD276), and HLA-G5, the content of which depends on the gestation ages [86]. Identification of these immunoregulatory proteins in exosomes confirms that these nanovesicles prevent trophoblast destruction by inducing maternal T cell apoptosis. These processes are likely to be responsible for pregnancy pathologies, such as preeclampsia (PE), fetal rejection, and fetal growth restriction [86].

Placental exosomes can participate in the control of gene expression by the miRNA MIR517 family. miR-517a-3p is transported by placental exosomes to the NK cells of pregnant women and reduces the expression of cGMP-dependent protein kinase PRKG1 [17]. It is assumed that miR-517a-3p is involved in regulating the activation and proliferation of maternal immune cells by inhibiting the NO/cGMP/PRKG1 pathway, probably contributing to maternal immune tolerance to the fetus. After the delivery, the concentration of miR-517a-3p significantly decreases in the NK cells of women, and in the cells of non-pregnant women is not detected. It is also suggested that miR-517a-3p of the placental exosomes is a potent activator NF-κB, likely having a significant influence on the immune response [17].

Trophoblast-derived exosomes increase monocyte migration in a dose-dependent manner and significantly enhance the production of IL-1beta, IL-6, serpin-E1, granulocyte colony-stimulating factor, granulocyte/monocyte colony-stimulating factor, and TNF-α [87]. These exosomes can attract monocytes and help them differentiate into tissue macrophages that support the trophoblast and secrete cytokines and chemokines necessary for the growth and survival of the trophoblast [87].

Extravillous trophoblast-derived exosomes and exosomes from pregnant women’s blood may contribute to endothelial cell migration [14,18,58]. In in vitro experiments, exosomes from maternal circulation at various gestational ages increased the migration of endothelial cells: exosomes from first trimesters by 2.7-fold, exosomes from the second trimester by 2.3-fold, and exosomes from the third trimester by 1.87-fold compared to cell migration without exosomes [14]. Researchers suggest a possible role of trophoblast-derived exosomes in the remodeling of the endometrium’s spiral arteries during pregnancy to ensure an adequate exchange of gases and nutrients, contributing to the successful development of the fetus [17,88]. In another study, maternal third-trimester exosomes were shown to significantly increase proliferation, migration, and tube formation of endothelial cells [58].

miRNA of C19MC cluster genes are packaged within trophoblast exosomes from trophoblast cells and attenuate viral replication in recipient cells: unlike non-placental cells, trophoblast cells are resistant to viral infections caused by coxsackievirus B3, poliovirus, vaccinia virus, cytomegalovirus, and vesicular stomatitis virus [71,89,90]. When non-placental cells of the lines U2OS, HFF, Huh7.5, and HeLa are cultured in a culture medium in which trophoblast cells have developed, the ability to resist infections is transferred from the trophoblast cells to the non-placental ones. miR-517-3p, miR-516b-5p, and miR-512-3p were shown to be transported by trophoblast exosomes to recipient cells and cause viral resistance and cell autophagy [89].

Given the COVID-19 pandemic, it is of interest to study trophoblast-derived exosomes’ antiviral activity against SARS-CoV-2 coronavirus since these exosomes are known to have antiviral activity [71,89,90]. Currently, it is known that the virus enters the human placenta: not only RNA of the virus is detected in the placenta [91], but also virions [92], while most newborns are healthy [93].

It should be noted that placental exosomes may have therapeutic potential. Exosomes obtained from the placental mesenchymal stromal cells induce the fusion of muscle cells, their differentiation, and an increase of expression of fibrogenic genes in myoblasts of patients with Duchenne muscular dystrophy is observed [94]. Exosomes of trophoblast and exosomes of placental mesenchymal stem cells have angiogenic activity [18,95]. Thus, exosomes can treat ischemic diseases, for example, in preeclampsia (PE).

The variety of proteins and miRNAs found in exosomes raises questions about whether proteins co-isolated with vesicles and exosomes’ proteins play an essential role in placental exosomes’ biological functions. However, it cannot be excluded that some of the physiological processes attributed to exosomes are functions of proteins that are co-isolated with them.

## 6. Placental Exosomes as Biomarkers of Pregnancy Pathologies

Exosomes have a tissue-specific composition, with changes in the concentration and composition of exosomes possibly associated with the pathological state of the placenta. Therefore, placental exosomes can become promising biomarkers of various early-stage pregnancy pathologies, such as PE and gestational diabetes mellitus.

### 6.1. Preeclampsia

Preeclampsia is a common and the most dangerous complication of pregnancy. PE and its associated hypertensive conditions account for 40% of preterm births [96]. The possibility of preventing and correcting PE is limited because its pathogenesis is still unclear. During normal placentation, the extravillous trophoblast invades the decidua and the spiral arteries of the endometrium [97]. As a result, trophoblast cells replace endothelial cells, leading to the expansion of arteries, and this remodeling of the spiral arteries ensures the adequate perfusion of the placenta. Extravillous trophoblast invasion in PE decreases, leading to placental hypoxia [97]. Placental exosomes from the blood of pregnant women with PE are currently being actively investigated because of their concentration and composition change during this pathology. The data on changes in the biochemical composition of placenta exosomes in PE are shown in Table 2.

The concentration of PLAP^+^-exosomes is increased in early-onset PE (<33 weeks) compared to normal pregnancy (<33 weeks). However, the content of PLAP^+^-exosomes in blood plasma decreases in late-onset PE (>34 weeks) compared to normal pregnancy (>34 weeks) [23,103].

The expression of exosomal miR-210 significantly increases in the blood of women with early-onset PE (31 weeks) compared to normal pregnant women [98]. It has been shown that this miRNA may reduce the invasion of trophoblast [104]. The content of miR-486-1-5p and miR-486-2-5p was significantly upregulated in the blood of women with PE compared to normal pregnant women, independent of gestational ages [99]. Expression of miR-155 was increased in placenta-associated serum exosomes from patients with PE (>34 weeks) compared with those from normal pregnant women [100]. Exosomes from the blood of pregnant women may inhibit eNOS expression in endothelial cells during PE development. This may be partly due to increased miR-155 expression in placenta-associated serum exosomes [100]. In this complication of pregnancy, the content of miR-141 is also enriched in the trophoblast exosomes [105]. This miRNA is involved in the proliferation and invasion of trophoblast cells. The miRNA-141 is transported by exosomes to T cells and reduces their proliferation, so it is assumed that placenta exosomes are involved in regulating normal and pathological pregnancy.

In peripheral blood of pregnant women in early-onset (<20 weeks) and late-onset (≥20 weeks) PE, exosomal miR-495, miR-494, and miR-136 content was 2.1-, 3.9-, and 6.4-times higher, respectively, compared to healthy women during early normal and late pregnancy [101]. These miRNAs can participate in the pathogenesis of PE by inhibiting cell proliferation and cell apoptosis. The authors suggest that miRNA-136, miRNA-494, and miRNA-495 of peripheral blood-derived exosomes could be used as potential non-invasive biomarkers in the early prediction of PE. These miRNAs can participate in PE pathogenesis by inhibiting cell proliferation and cell apoptosis [101].

Comparing the lipid composition of syncytiotrophoblast exosomes in normal and pathological pregnancy revealed significant lipid content changes in patients with PE [102]. The phosphatidylserine level was significantly upregulated, while phosphatidylinositol, phosphatidic acid, and ganglioside mannoside 3 were downregulated considerably in PE patients compared to normal pregnancy women [102]. The content of sphingomyelin SM-18:0 was increased in exosomes from the maternal circulation in early-onset PE [57]. Then the truncated endoglin was released into the maternal circulation via sphingomyelin SM-18:0-enriched exosomes together with TGF-beta receptors 1 and 2. Such an exosomal TGF-beta receptor complex could be functionally active and block the vascular effects of TGF-beta in PE women [57].

Over 400 proteins were identified in placenta syncytiotrophoblast exosomes with 25 proteins being differentially expressed in PE compared with healthy pregnant controls, including integrins, annexins, and histones [106]. The composition of proteins and lipids is likely to be of great importance in implementing immune responses, regulating vascular tone and blood clotting, and developing oxidative stress and apoptosis.

Exosomes from maternal circulation express soluble Fms-like tyrosine kinase-1 (sFlt-1) and soluble endoglin (sEng), thus increasing vasoconstriction [19]. In PE, the content of these proteins is upregulated. In vitro, these exosomes may attenuate the proliferation, migration, and tube formation of human umbilical vein endothelial cells [19]. In a mouse model, an injection of PE exosomes from the plasma of women resulted in decreased body weight and elevated blood pressure of pregnant mice, with the fetus size and weight also reduced compared with the control group (an injection of exosomes from normal pregnancy) [19].

### 6.2. Gestational Diabetes Mellitus

Gestational diabetes mellitus (GDM) is known to affect approximately 6% of pregnant women, and prevalence is increasing in parallel with the obesity epidemic [107]. Hyperglycemia can significantly affect the concentration of exosomes as well as their biochemical composition [83,84]. The data on changes in the biochemical composition of human placenta exosomes in gestational diabetes mellitus (GDM) are shown in Table 3.

The concentration of placental exosomes in the blood of pregnant women is higher with GDM compared to normal pregnant women [109]: in the first trimester, the concentration of PLAP^+^-exosomes was 1.6-fold higher (81 ± 7 vs. 128 ± 14 pg/mL for normal and GDM, respectively), in the second trimester it was 1.5-fold higher (188 ± 14 vs. 282 ± 24 pg/mL for normal and GDM, respectively), and the third trimester it was 1.3-fold higher (304 ± 29 vs. 418 ± 57 pg/mL for normal and GDM, respectively).

In GDM, the contents of 9 miRNAs in trophoblast exosomes were significantly increased, and the ranges of 14 miRNAs were decreased [74]. Exosomes isolated from GDM pregnancies significantly increased the release of proinflammatory cytokines (GM-CSF, IL-4, IL-6, IL-8, IFN-γ, and TNF-α) by endothelial cells. The dysregulation of exosome effects and/or function on endothelial cells may be implicated in the proinflammatory state of GDM. 

The level of active dipeptidyl peptidase-4 increases 8-fold in placental exosomes in pregnant women with GDM in the late pregnancy stage [108]. Dipeptidyl peptidase-4 cleaves the glucagon-like polypeptide-1, which increases insulin secretion so that placental exosomes may be involved in the pathogenesis of gestational diabetes mellitus.

Placental exosomes from pregnant patients with GDM have differential expression of 78 proteins, including spectrin alpha erythrocytic 1, CAMK2β, PAPP-A, perilipin 4, fatty acid-binding protein 4, and hexokinase-3 compared to normal pregnant women [24]. The following differences have been observed in the protein abundance of PAPP-A and CAMK2β within exosomes isolated from compared with GDM by ELISA: PAPP-A was downregulated and CAMK2β was upregulated in exosomes from pregnant women with GDM.

Thus, it is possible that over time the assessment of changes in the content and composition of exosomes may form the basis for determining potential diagnostic markers of the pathogenesis of specific pregnancy pathologies.

## 7. Conclusions

Exosomes contain various proteins, RNA, and lipids, which can play a crucial role in implementing the biological functions of these nanovesicles [110,111]. Human placenta exosomes obtained from the blood of pregnant women may be used as biomarkers of early pregnancy pathological course in patients without symptoms [19,24,98,99,100,101,108]. To use placental exosomes for diagnostic purposes, it is necessary to carefully study their structure, functions, and physiological characteristics [44,51,52,62]. Most studies currently use a mixture of all extracellular vesicles contaminated with co-isolating proteins, protein complexes, and lipoproteins. Despite significant progress in searching for ways to purify exosomes from various impurities, no single correct protocol can be used to quickly, efficiently, and economically obtain homogeneous samples of these nanovesicles. Many studies have shown that all the approaches listed above make it possible to obtain preparations only enriched with exosomes but not homogeneous exosome preparations [44,51,62]. The question of exosome intrinsic components and their possible biological functions remains open.

## Figures and Tables

**Figure 1 ijms-22-02158-f001:**
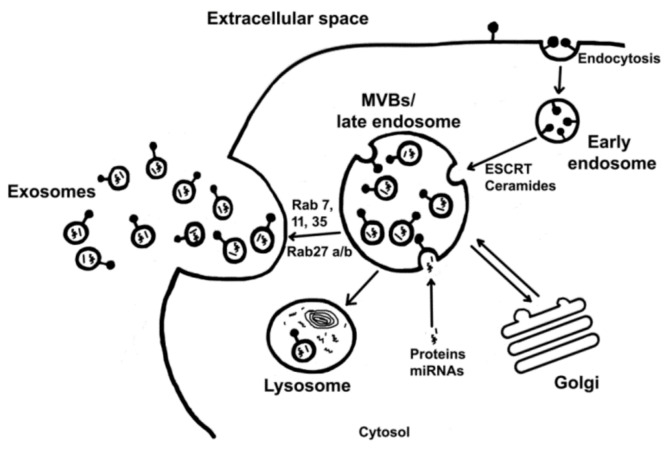
Schematic representation of exosome biogenesis and secretion. The process leading to exosome secretion can be divided into three steps: (1) exosome biogenesis, (2) transport of MVBs (multivesicular bodies) to the plasma membrane, and (3) fusion of MVBs with the plasma membrane.

**Figure 2 ijms-22-02158-f002:**
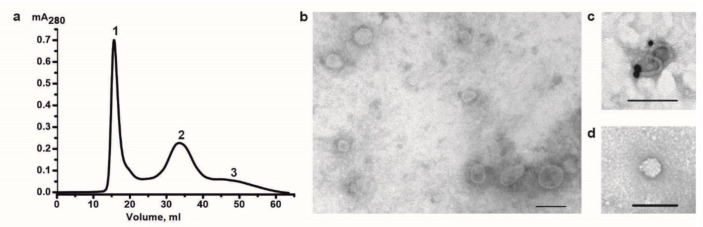
(**a**) Size-exclusion chromatography of crude exosome preparations on Sepharose 4B. Crude exosome preparations from human placenta were obtained by several differential centrifugation and ultrafiltration through a 0.1 μm filter (—), absorbance at 280 nm (A_280_). (**b–d**) Structural components of exosome preparations after size-exclusion chromatography: vesicles with different size (**b**); CD63^+^-exosomes (**c**); “non-vesicles” (**d**). Transmission electron microscopy, negative staining. The length of the scale bar corresponds to 100 nm.

**Table 1 ijms-22-02158-t001:** The most represented proteins of trophoblast-derived exosomes.

Most Represented Proteins	Number of Proteins Described	Exosome Source/Ref	Isolation Method	Exosome Markers
PLAP, serotransferrin, annexin A5, α- and β-subunits of hemoglobin, serum albumin, β-actin, CD81	11–13 ^1^	Placentas after term delivery [51]	Combination of differential centrifugations, gel filtration, ultrafiltration (0,1 μm), and affinity chromatography	CD63, CD81, PLAP
β-actin, albumin, α-2- macroglobulin, apolipoproteins A and M, fibronectin 1, enolase, lactate dehydrogenase B	140 ^2^	Cell lines JEG-3 (choriocarcinoma) and HTR-8/SVneo (from 6–12 weeks the first-trimester placenta) [18]	Differential centrifugations + sucrose cushion ultracentrifugation	CD63
Actin, α-actinin-1, cofilin, filamin-A-B-C, ICAM-1, tubulins, gelsolin, profilin-1, spectrin, simplekin, mucin-4, talin, vinculin, myosins, α3-, β1- and αv- integrins	282 ^2^	Cell line Sw71 (from 7 weeks first-trimester placenta) [54]	Differential centrifugations + sucrose density gradient ultracentrifugation	CD81, ALIX
PLAP, proteins involved hematopoiesis, primary immunodeficiency signaling and B cell development	340 ^2^	Blood of pregnant women (6–12 weeks) [13]	Differential centrifugations + sucrose density gradient ultracentrifugation	CD63, PLAP
Proteins involved in vesicular transport and inflammation	1476 ^2^	First trimester (8–12 weeks) placental explant [56]	Differential centrifugations + iodixanol density gradient ultracentrifugation	CD63
PLAP, ADAM10, Integrin β4, integrin α6, annexins A2 and A6, syntenin-1, catenin α1, calpain-6, choline transporter-like protein 2, TNF α-induced protein 3	1684 ^2^	Trophoblast cells from placentas after term pregnancy and delivery [69]	Differential centrifugations + iodixanol density gradient ultracentrifugation	CD63

^1^ MALDI-TOF-MS/MS of tryptic hydrolysates after 2D-electrophoresis. ^2^ LC-MS/MS of tryptic hydrolysates.

**Table 2 ijms-22-02158-t002:** Changes in levels of proteins, miRNAs, and lipids in placental exosomes in preeclampsia.

Exosomes Source/Ref	Isolation Method	Targets	Detection Method
Plasma [98]	Commercial kit ExoRNEasy	miR-210 ↑	RT-PCR
Plasma [99]	Differential centrifugations + sucrose density gradient ultracentrifugation	miR-486-1-5p ↑ miR-486-2-5p ↑	RT-PCR
Serum [100]	Differential centrifugations + sucrose density gradient ultracentrifugation	miR-155 ↑	RT-PCR
Plasma [101]	Differential centrifugations	miR-136 ↑ miR-494 ↑ miR-495 ↑	RT-PCR
Placental villi [102]	Differential centrifugations	Phosphatidylserine ↑ Phosphatidylglycerol ↓ Ganglioside mannoside 3 ↓	HPLC/MS
Plasma [57]	Commercial kit ExoQuick + sucrose density gradient ultracentrifugation	Sphingomyelin SM 18:0 ↑	MALDI-MSI
Plasma [19]	Commercial kit ExoQuick	sFlt-1 ↑, sEng ↑	ELISA

↑—upregulated; ↓—downregulated.

**Table 3 ijms-22-02158-t003:** Changes in levels of proteins and miRNAs in placental exosomes in gestational diabetes mellitus.

Exosomes Source/Ref	Isolation Method	Targets	Detection Method
Explants of chorionic villi [74]	Differential centrifugations + iodixanol density gradient ultracentrifugation	miR-125a-3p ↑, miR-224-5p ↑, miR-584-5p ↑, miR-186-5p ↑, miR-22-3p ↑, miR-99b-5p ↑, miR-433-3p ↑, miR-197-3p ↑, miR-423-3p ↑ miR-208a-3p ↓, miR-335-5p ↓, miR-451a ↓, miR-145-3p ↓, miR-369-3p ↓, miR-483-3p ↓, miR-203a-3b ↓, miR-574-3p ↓, miR-144-3p ↓ miR-6795-5p ↓, miR-550a-3-3p ↓, miR-411-5p ↓, miR-550a-3-3p ↓, miR-140-3p ↓	RT-PCR
Placental perfusate, Plasma [108]	Differential centrifugations	Dipeptidyl peptidase IV ↑	Flow cytometry
Plasma [24]	Differential centrifugations + size-exclusion chromatography	CAMK2β ↑, PAAP-A ↓ and other 76 proteins dysregulated	SWATH-MS, ELISA

↑—upregulated; ↓—downregulated.

## Data Availability

The data that supports the findings of this study are available within the article and its Appendix A.

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
