# Peer review of "Human Placenta Exosomes: Biogenesis, Isolation, Composition, and Prospects for Use in Diagnostics"

_ijms, 2021, doi:10.3390/ijms22042158_

Round 1

Reviewer 1 Report

To authors,

The theme is important. The authors incorporated my advice into the version. English has become better. Although the cause-effect relationship of placenta exosome and placental pathologies are still obscure, the authors should not be blamed for it. The data/review will contribute to the future study on this issue.

Author Response

Reply to reviewer 1.

To authors,

The theme is important. The authors incorporated my advice into the version. English has become better. Although the cause-effect relationship of placenta exosome and placental pathologies are still obscure, the authors should not be blamed for it. The data/review will contribute to the future study on this issue.

Answer: Thank you very much for the analysis of the paper.

Reviewer 2 Report

In this new submission of a previously submitted manuscript (ijms-1003037), the authors have satisfactorily responded to all my comments. However, upon further read, I have a few additional comments: 

  • The authors have thoroughly argued that the most impeding factor preventing the use of exosomes as biomarkers for health and disease in pregnancies is the absence of an efficient and reproducible exosome isolation technique. I am wondering what are the authors suggestions' as future directions in this regard?

  • After the authors assessment of the literature, do they really want to limit the size range of exosomes to 40-100 nm (lines 10 and 28)?

  • The sentence line 29 need to be rewritten. An electron microscope does not determine the floating density of exosomes. 

  • There is a grammatical error line 176: missing "and" before "trophoblast". The sentence would also read better if "after term delivery" was moved to before "placental perfusate".  

Other than that, the manuscript was really well-written and can make a great reference for scientists studying exosomes in pregnancies. 

Author Response

Reply to reviewer 2.

Comments and Suggestions for Authors

In this new submission of a previously submitted manuscript (ijms-1003037), the authors have satisfactorily responded to all my comments. However, upon further read, I have a few additional comments: 

  • The authors have thoroughly argued that the most impeding factor preventing the use of exosomes as biomarkers for health and disease in pregnancies is the absence of an efficient and reproducible exosome isolation technique. I am wondering what are the authors suggestions' as future directions in this regard?

Answer: Exosomes from different sources are usually isolated using standard methods – centrifugation, ultrafiltration, and ultracentrifugations. Exosomes isolated by these procedures contain from a few dozen to thousands of different proteins [Front Pharmacol. 2014, 5, 175; Hum Reprod. 2016, 31, 687–699; Placenta 2016, 47, 86–95]. To obtain the most purified exosome preparations, it is necessary to use a combination of several methods – gel filtration, affinity chromatography using antibodies to the main exosomal proteins. Using electron microscopy, it is necessary to analyze the homogeneity of the preparations. According to the paper of Grigor’ieva A.E. et al. [Biomeditsinskaya khimiya, 2016, 62, 99–106.], the proportion of “non-vesicles” in the preparation of exosomes can be up to 40%. “Non-vesicles” correspond in morphology to lipoproteins of different density. “Non-vesicles” co-isolated with these nanovesicles may also carry disease markers and molecular signals. Furthermore, the functions that are attributed to exosomes may not belong to them at all. Therefore, it should be necessary to try to separate “non-vesicles” (lipoproteins) from exosomes using affinity chromatography using antibodies to the lipoprotein proteins.

  • After the authors assessment of the literature, do they really want to limit the size range of exosomes to 40-100 nm (lines 10 and 28)?

Answer: Exosomes are nanovesicles of endosomal origin, formed by the inward budding multivesicular bodies (MVB). When studied in the electron microscope, the intraluminal vesicles in MVB have size 50–90 nm [J Cell Biol. 1985, 101, 942-948; Am J Reprod Immunol. 2010, 6, 520-533.].

Exosomes are enriched with late endosomal membrane markers: CD63, CD9, and CD81. It was shown using the immunoelectron microscopy that isolated extracellular vesicles with these tetraspanins on their surface (CD63, CD9, and CD81) have size 40–100 nm [Biomeditsinskaya khimiya, 2016, 62, 99–106]. Large vesicles (with size more than 100 nm) do not have these tetraspanins on the surface.

  • The sentence line 29 need to be rewritten. An electron microscope does not determine the floating density of exosomes.

Answer: corrected, please see lines 27-28

  • There is a grammatical error line 176: missing "and" before "trophoblast". The sentence would also read better if "after term delivery" was moved to before "placental perfusate".

Answer: corrected, please see lines 170-171

Other than that, the manuscript was really well-written and can make a great reference for scientists studying exosomes in pregnancies.

Answer: Thank you very much for the analysis of the paper.

Reviewer 3 Report

The presented text is significantly better written than before, although it still requires English language corrections by English native speaker, especially chapter Conclusions.

as well as:

  • line 146 many methods are used exosome to isolate exosomes from placenta
  • line 468 IL-1beta instead of IL-1b
  • line 511 - please specify the PE abbreviation;
  • line 586 & 587 should be: were significantly increased; were decreased; 

please localize SFig2 in the text of article. 

Author Response

Reply to reviewer 3.

The presented text is significantly better written than before, although it still requires English language corrections by English native speaker, especially chapter Conclusions.

as well as:

  • line 146 many methods are used exosometo isolate exosomes from placenta

Answer: corrected, please see line 148

  • line 468 IL-1beta instead of IL-1b

Answer: corrected, please see line 479

  • line 511 - please specify the PE abbreviation;

Answer: corrected, please see line 513

  • line 586 & 587 should be: were significantly increased; were decreased; 

Answer: corrected, please see lines 600-601

  • please localize SFig2 in the text of article.

Answer: corrected, please see Fig.2 in the text.

Reviewer 4 Report

---

Author Response

Thank you very much for the analysis of the paper.

This manuscript is a resubmission of an earlier submission. The following is a list of the peer review reports and author responses from that submission.

Round 1

Reviewer 1 Report

The authors have submitted a review manuscript on human placental exosomes. The authors have structured their review into biogenesis, isolation, composition and use in diagnostics of exosomes. The structure is good; however, in most of the chapters the placental aspect is more or less completely missing. This is outlined in “major concerns”.

The authors have used this review to publish detailed data from their own studies. This should be completely avoided as this is a review that should give a balanced view on the topic rather than focusing on own data (again, see “major concerns”).

A huge number of specific concerns have been identified all of which show the work that needs to be put into this manuscript.

General concern

  1. Proof reading of a native speaker is recommended.
  2. When describing “placental exosomes” the authors need to be more precise as the placenta contains quite a number of different cell types. Placental exosomes also include exosomes from placental endothelial cells communicating with the embryo/fetus. So please always define cell type and gestational age (or trimester) of collection.

Major concerns

  1. The authors state to have submitted a manuscript on human placental exosomes. However, 95% of what the authors describe is general information on exosomes. As an example: In chapter 2 (Biogenesis and secretion of exosomes, lines 63-144) there is a single sentence (lines 138-140) describing information on placental exosomes. Hence, the authors need to change the title or need to better focus on placental, specifically trophoblast-derived exosomes. The focus on the placenta would be much more valuable as the topic of placental exosomes is still underrepresented.
  2. Similarly, chapter 3 on the isolation of exosomes starts with a general sentence on the isolation of placental exosomes. However, all the methods listed thereafter are general protocols not focusing on placental exosomes.
  3. 200-277: This whole part of chapter 3 on the isolation of exosomes is a description of a couple of studies from the authors. There are a lot of details with three figures that should be presented in an original articles but not in such a review. Hence, the only description of isolation methods for placental exosomes is based on studies of the authors. This part needs to be deleted completely including figures 2, 3 and 4, and replaced by a much wider description and discussion of isolation methods of placental exosomes.
    This is also true for lines 340-369 and figure 5!
  4. Some figures in this manuscript are identical with already published images from the same group! This reader compared the figures of this manuscript with only two recent publications of the same group:
    [60] Burkova EE et al. IJMS 2019;20:2434.
    [63] Burkova EE et al. IUBMB Life 2018;70:1144-55.

- Figure 2A is identical with figure 2B in [60].

- Figure 2C is identical with figure 2D in [60].

- Figure 2D is identical with the inset in figure 2A in [60].

- Figure 3B is identical with figure 3B in [60].

- Figure 3C is identical with figure 3G (turned 180°) in [60].

- Figure 3D is identical with a part of figure 3E (and showing increased light intensity) in [60].

- Figure 4C is identical with figure 2F in [63].

- Figure 4E is identical with figure 3I in [63].

- Figure 4F is identical with a part of figure 3E in [63].

Specific concerns

153-159: It is the opinion of the authors that electron microscopy is the best method to analyze exosomes. Other scientists may have a different view. So please rewrite phrases such as “Immunoelectron microscopy is the best-known method” and “Transmission electron microscopy is also the best method”. Such phrases need to be toned down.

289-290: “analyzed on insufficiently purified exosome preparations.” The authors should be cautious here! An isolation of a biological structure is never 100% pure. Who defines when an isolation procedure results in an insufficiently poor isolate? Thus, tone down the wording here! Of course, the different isolation procedures will result in a varying degree of “purity” and thus will have a direct impact on the results and thus the comparability of such results. What is needed is a much better description of what has been isolated rather than statements about insufficient purity.

Table 1: This table nicely shows the problems with placental exosomes. Not a single study listed in this table has been performed with intact villous tissues (villous explant culture or perfused placenta). Studies listed in this table are using placental homogenates (so no isolation of exosomes can be performed), cell lines not representing placental trophoblast such as HTR-8/SVneo and Swan71 and isolated primary trophoblasts. Please add “real” exosome isolations if possible.

303-304: PLAP is not specific to the placenta in general, but rather specific to the syncytiotrophoblast. Please specify.

304-308: The authors need to be more specific! Please define from which cell type the “placental exosomes” were derived.

377-378: Again, electron microscopy is not the one and only and always the best method. Please rephrase.

394-396: Please better specify which cells are described here.

401-402: Again, specify cell types. What are “chorionic villi cells”?

450-452: Please define what is hypoxia for a first trimester trophoblast cell. These cells live under normoxic conditions with a pO2 of about 20mmHg.

459-465: With all the uncertainty of exosome isolations, now the authors clearly state that 12% to 25% of the exosomes in maternal blood during pregnancy are placenta-derived. The authors also claim that only placenta-derived exosomes increase the release of factors from endothelial cells. How do the authors know?

470-471: Are the syncytin receptors present on other cells than BeWo cells? BeWo cells are derived from a choriocarcinoma and hence are also derived from trophoblast.

472-484: Is it always clear that the data presented here are derived from placental (i.e. trophoblast) exosomes? How were these exosomes separated from maternal exosomes?

489-490: This reader would guess that only exosomes from extravillous trophoblasts carry HLA-G5. Please define.

519-522: How can exosomes help in reconstructing the walls of spiral arteries? There is no reference to this statement.

Table 2: How can exosomes from cell lines be listed in a table that deals with placental exosomes in preeclampsia. Please focus on maternal blood derived exosomes or exosomes derived from villous explant cultures or placental perfusates of preeclamptic placentas!

565-566: Please define gestational age and type of preeclampsia!

573-574: What are “trophoblast T cells”?

576-578: Please define gestational age and type of preeclampsia!

Author Response

Reviewer 1

The authors have submitted a review manuscript on human placental exosomes. The authors have structured their review into biogenesis, isolation, composition and use in diagnostics of exosomes. The structure is good; however, in most of the chapters the placental aspect is more or less completely missing. This is outlined in “major concerns”.

The authors have used this review to publish detailed data from their own studies. This should be completely avoided as this is a review that should give a balanced view on the topic rather than focusing on own data (again, see “major concerns”).

A huge number of specific concerns have been identified all of which show the work that needs to be put into this manuscript.

General concern

  1. Proof reading of a native speaker is recommended.

Answer: It was impossible to do the language editing in 10 days, but we will do it in the next round.

  1. When describing “placental exosomes” the authors need to be more precise as the placenta contains quite a number of different cell types. Placental exosomes also include exosomes from placental endothelial cells communicating with the embryo/fetus. So please always define cell type and gestational age (or trimester) of collection.

Answer: We added information about cell types, gestational ages. Please see Table 1; lines  260-261; 263-264; 275; 279; 334-336; 386; 413-414; 498

Major concerns

  1. The authors state to have submitted a manuscript on human placental exosomes. However, 95% of what the authors describe is general information on exosomes. As an example: In chapter 2 (Biogenesis and secretion of exosomes, lines 63-144) there is a single sentence (lines 138-140) describing information on placental exosomes. Hence, the authors need to change the title or need to better focus on placental, specifically trophoblast-derived exosomes. The focus on the placenta would be much more valuable as the topic of placental exosomes is still underrepresented.

Answer: We clarified the text; please see lines 70-76; 85-86; 89-92; 127-129. Unfortunately, at present, there is very little information about the biogenesis of placental exosomes. The placental exosome biogenesis has been studied mainly at the ultrastructural level using MICA/B, ULBP 1–5, FasL, TRAIL, PD-L1, also the specific exosomal markers the biogenesis.

  1. Similarly, chapter 3 on the isolation of exosomes starts with a general sentence on the isolation of placental exosomes. However, all the methods listed thereafter are general protocols not focusing on placental exosomes.

Answer: We clarified the text; please see lines 156-159; 162-164; 166-168; 176-181; 182-185; 191-200.

  1. 200-277: This whole part of chapter 3 on the isolation of exosomes is a description of a couple of studies from the authors. There are a lot of details with three figures that should be presented in an original articles but not in such a review. Hence, the only description of isolation methods for placental exosomes is based on studies of the authors. This part needs to be deleted completely including figures 2, 3 and 4, and replaced by a much wider description and discussion of isolation methods of placental exosomes.
    This is also true for lines 340-369 and figure 5!

Answer: We changed the figures that were published before and changed them for the new ones. Also, we moved these figures to the Supplementary materials (please see Supplementary file materials).

These figures are necessary to show that the isolation of exosomes using the standard protocol leads to the isolation of exosome preparations contaminated with various co-isolating proteins, non-exosomal vesicles, and non-vesicles. According to the paper of Grigor’ieva A.E. et al. [Biomeditsinskaya khimiya, 2016, 62, 99–106.], the proportion of impurities in the preparation of exosomes can be up to 40%, and these co-isolating structures can exhibit various biological activities. Thus the functions that are attributed to exosomes may not belong to them at all. Using TEM microphotographs, we want to draw readers' attention to the importance of analyzing the homogeneity of exosome preparations.

  1. Some figures in this manuscript are identical with already published images from the same group! This reader compared the figures of this manuscript with only two recent publications of the same group:
    [60] Burkova EE et al. IJMS 2019;20:2434.
    [63] Burkova EE et al. IUBMB Life 2018;70:1144-55.

- Figure 2A is identical with figure 2B in [60].

- Figure 2C is identical with figure 2D in [60].

- Figure 2D is identical with the inset in figure 2A in [60].

- Figure 3B is identical with figure 3B in [60].

- Figure 3C is identical with figure 3G (turned 180°) in [60].

- Figure 3D is identical with a part of figure 3E (and showing increased light intensity) in [60].

- Figure 4C is identical with figure 2F in [63].

- Figure 4E is identical with figure 3I in [63].

- Figure 4F is identical with a part of figure 3E in [63].

Answer: We have replaced these figures with previously unpublished ones. Please see Supplementary materials.

Specific concerns

153-159: It is the opinion of the authors that electron microscopy is the best method to analyze exosomes. Other scientists may have a different view. So please rewrite phrases such as “Immunoelectron microscopy is the best-known method” and “Transmission electron microscopy is also the best method”. Such phrases need to be toned down.

Answer: We changed the sentences (please see lines 143–145). Our opinion is that an essential task of exosome isolation is analyzing their morphology and homogeneity using TEM. Since exosomes, other vesicles, protein aggregates have the size from tens to hundreds of nanometers, it is impossible to analyze their homogeneity with the naked eye. Immunoelectron microscopy allows detecting tetraspanins DIRECTLY on the surface of vesicles. Methods of western blot and flow cytometry fix all particles in the solution with specific exosomal markers, so there is no selectivity for analyzing membrane and non-membrane structures. Method of NTA allows getting only estimated parameters – the hydrodynamic size of particles that can exceed the physical size several times. Also, the device analyzes all the particles and their aggregates that are present in the solution.

289-290: “analyzed on insufficiently purified exosome preparations.” The authors should be cautious here! An isolation of a biological structure is never 100% pure. Who defines when an isolation procedure results in an insufficiently poor isolate? Thus, tone down the wording here! Of course, the different isolation procedures will result in a varying degree of “purity” and thus will have a direct impact on the results and thus the comparability of such results. What is needed is a much better description of what has been isolated rather than statements about insufficient purity.

Answer: We rephrased this statement; please see lines 242–244.

Table 1: This table nicely shows the problems with placental exosomes. Not a single study listed in this table has been performed with intact villous tissues (villous explant culture or perfused placenta). Studies listed in this table are using placental homogenates (so no isolation of exosomes can be performed), cell lines not representing placental trophoblast such as HTR-8/SVneo and Swan71 and isolated primary trophoblasts. Please add “real” exosome isolations if possible.

Answer: The placenta formation starts at the 6th week of pregnancy. The trophoblast is differentiated in two ways during its growth – villous and extravillous (please see refs 9). Thus, the extra-vascular trophoblast is the part of the placenta. Cells HTR-8/SVneo were isolated from 6-12 weeks, and cell Swan71 were isolated from 7 weeks of the first-trimester placenta from a normal pregnancy. Also, these cells were immortalized by SV40. Both cell lines from the extravillous cytotrophoblast originate from the extravillous trophoblast. Cells HTR-8/SVneo and Swan71 are used in many works. The studies using first-trimester trophoblast cells may be limited by the inability to obtain patient samples and adequate cell numbers; both cell lines are frequently used as models of physiologically invasive extravillous trophoblast. In many studies, it was shown that cells HTR-8/SVneo and Swan71 have the placental phenotype; for example, HTR-8/SVneo expressed placental alkaline phosphatase (PLAP) [Placenta 2009, 30, 939–948; Placenta 2011, 32, 771–777].

Moreover, many studies on "placental exosomes" are conducted on exosomes isolated from trophoblast cells obtained from the placenta at different pregnancy stages, including these immortalized cells [Front Pharmacol. 2014, 5, 175; Am J Obstet Gynecol. 2015, 213, S173-181]. Such exosomes derived from immortalized trophoblast were named "placental". This review shows that proteins, nucleic acids, and possible biological functions are performed by certain placental exosomes that can be secreted by various cells throughout the placenta's existence.

We modified Table 1 and clarified the text (please see lines 238-244; 301-311)

Using this table 303-304: PLAP is not specific to the placenta in general, but rather specific to the syncytiotrophoblast. Please specify.

Answer: We specified the details (please see line 259-260 )

304-308: The authors need to be more specific! Please define from which cell type the “placental exosomes” were derived.

Answer: We added this information. Please see lines 260-264; 275; 279.

377-378: Again, electron microscopy is not the one and only and always the best method. Please rephrase.

Answer: We changed the sentence. Please see lines 318-319.

394-396: Please better specify which cells are described here.

Answer: We specified these cells. Please see lines 334-336.

401-402: Again, specify cell types. What are “chorionic villi cells”?

Answer: We clarified the text; please see lines 341-342.

450-452: Please define what is hypoxia for a first trimester trophoblast cell. These cells live under normoxic conditions with a pO2 of about 20mmHg.

Answer: We added this information. Please see lines 386–388.

459-465: With all the uncertainty of exosome isolations, now the authors clearly state that 12% to 25% of the exosomes in maternal blood during pregnancy are placenta-derived. The authors also claim that only placenta-derived exosomes increase the release of factors from endothelial cells. How do the authors know?

Answer: We clarified this text; please see lines 394-397. We describe the paper [Placenta 2017, 50, 60–69]. In this paper, total plasma exosomes were isolated using ultracentrifugation 100000×g and ultracentrifugation. The concentration of placental exosomes was quantified using PLAP by immunofluorescent NTA.

470-471: Are the syncytin receptors present on other cells than BeWo cells? BeWo cells are derived from a choriocarcinoma and hence are also derived from trophoblast.

Answer: Trophoblast cells have syncytin receptors. We clarified the text; please see lines 401-404.

472-484: Is it always clear that the data presented here are derived from placental (i.e. trophoblast) exosomes? How were these exosomes separated from maternal exosomes?

Answer: We clarified the text; please see lines 405-415.

489-490: This reader would guess that only exosomes from extravillous trophoblasts carry HLA-G5. Please define.

Answer: HLA-G5 is expressed mostly by trophoblast cells. In the paper [Placenta 2012, 33, 982–990] trophoblast-derived exosomes from early and term placentas can expressed this molecule HLA-G5. Please see lines 421-424.

519-522: How can exosomes help in reconstructing the walls of spiral arteries? There is no reference to this statement.

Answer: We clarified the text (please see 440-449) and added references (please see ref 87)

Table 2: How can exosomes from cell lines be listed in a table that deals with placental exosomes in preeclampsia. Please focus on maternal blood derived exosomes or exosomes derived from villous explant cultures or placental perfusates of preeclamptic placentas!

Answer: We clarified this table; please see Table 2.

565-566: Please define gestational age and type of preeclampsia!

Answer: We added this information; please see 492-497.

573-574: What are “trophoblast T cells”?

Answer: This is a typo. We deleted "trophoblast." Please see lines 502-504.

576-578: Please define gestational age and type of preeclampsia!

Answer: We added gestational age and type of preeclampsia. Please see lines 505-507.

Thank you very much for the critical analysis of the paper.

Reviewer 2 Report

To authors, 

The paper is well structured and well written. English is easy to read and comfortable. I wish you to consider the following two points.

  1. You, here and there, described preeclampsia (PE) and related placental exosome to this disorder (PE). Since the readers of this journal are not always specialists of PE, you had better shortly describe the pathophysiology of PE. In short, PE is characterized by inappropriate (possibly reduced) invasion of extravillous trophoblast (first-step) and this causes eventual hypoxia of the placenta (trophoblast). This, next, causes systemic endothelial dysfunction (second-step), which eventually causes PE manifestation. I mean the two-step theory of PE. In this review, you mainly targeted to the first step, which may be OK. The point is that PE is caused by inappropriate (reduced) invasion of trophoblast and thus, in this regard, exosome may participate in (have some association with) the pathogenesis of PE. There may be some other mechanisms/explanation for the relationship between exosome and PE. Anyhow, the mechanism of PE is a little complicated, and thus, you had better explain this fundamental “context” regarding the relation between exosome and PE. Adding this explanation may be reader-friendly.
  2. In my understanding, cause-effect relationship between the disease (for example PE) and exosome is not well established. We only partly now the “phenomenon”, “aspects”, or “facts” between the disease and exosome. We, irrespective of cause-effect relationship between the two, “utilizes” the phenomenon/fact/aspects to diagnose corresponding disorders. For readers’ better understanding, I believe that you had “touch” this issue. For the description of 1 and 2 (if you agree with me), please do not expand the volume any more. I only wish you to make the context clearer by “touching” these two. This may improve the paper quality, I believe.

Author Response

Reviewer 2

To authors, 

The paper is well structured and well written. English is easy to read and comfortable. I wish you to consider the following two points.

  1. You, here and there, described preeclampsia (PE) and related placental exosome to this disorder (PE). Since the readers of this journal are not always specialists of PE, you had better shortly describe the pathophysiology of PE. In short, PE is characterized by inappropriate (possibly reduced) invasion of extravillous trophoblast (first-step) and this causes eventual hypoxia of the placenta (trophoblast). This, next, causes systemic endothelial dysfunction (second-step), which eventually causes PE manifestation. I mean the two-step theory of PE. In this review, you mainly targeted to the first step, which may be OK. The point is that PE is caused by inappropriate (reduced) invasion of trophoblast and thus, in this regard, exosome may participate in (have some association with) the pathogenesis of PE. There may be some other mechanisms/explanation for the relationship between exosome and PE. Anyhow, the mechanism of PE is a little complicated, and thus, you had better explain this fundamental “context” regarding the relation between exosome and PE. Adding this explanation may be reader-friendly.

Answer: We added some discussion (please see lines 480-486)

  1. In my understanding, cause-effect relationship between the disease (for example PE) and exosome is not well established. We only partly now the “phenomenon”, “aspects”, or “facts” between the disease and exosome. We, irrespective of cause-effect relationship between the two, “utilizes” the phenomenon/fact/aspects to diagnose corresponding disorders. For readers’ better understanding, I believe that you had “touch” this issue. For the description of 1 and 2 (if you agree with me), please do not expand the volume any more. I only wish you to make the context clearer by “touching” these two. This may improve the paper quality, I believe

Answer: We added some discussion (please see lines 480-486; 500-504; 526-532)

Reviewer 3 Report

The review article submitted for evaluation is communicative and shows the current information about human placenta exosomes in the way expressed in the title.

The language should be improved, many repeats, (lines 17& 20; 39&42; 294 &294; 570&570 - because of //due to ), please eliminate several typos and punctuation errors ex. lines 39, 526

Please use the proper abbreviation for interferon IFN rather than INF (615), and Intraluminal vesicles which seems to be abbreviated ILVs instead of IVLs

The abbreviated words and phrases should be clarified in long form at every first time used in the text to have better understanding of the text - preeclampsia – PE line 515 rather than line 555.

line 604 - the number of Table 2 is incorrect, should be enumerated as Table 3

please write correctly the references in the Table 1.

Please correct the numbering of reference No 2 which appears latter than reference No10.

Author Response

Reviewer 3

The review article submitted for evaluation is communicative and shows the current information about human placenta exosomes in the way expressed in the title.

  1. The language should be improved, many repeats, (lines 17& 20; 39&42; 294 &294; 570&570 - because of //due to ), please eliminate several typos and punctuation errors ex. lines 39, 526

Answer: Corrected

  1. Please use the proper abbreviation for interferon IFN rather than INF (615), and Intraluminal vesicles which seems to be abbreviated ILVs instead of IVLs

Answer: Corrected.

  1. The abbreviated words and phrases should be clarified in long form at every first time used in the text to have better understanding of the text - preeclampsia – PE line 515 rather than line 555.

Answer: Corrected.

  1. line 604 - the number of Table 2 is incorrect, should be enumerated as Table 3

Answer: Corrected.

  1. please write correctly the references in the Table 1.

Answer: Corrected.

  1. Please correct the numbering of reference No 2 which appears latter than reference No10.

Answer: We corrected these references

Reviewer 4 Report

As stated in its title “Human placenta exosomes: biogenesis, isolation, composition, and prospects for use in diagnostics”, the present review by Burkova et al. discussed studies that focused on a subtype of extracellular vesicles secreted by the human placenta ––or  human placenta exosomes. The authors’ main message is that the field is still in its infancy since there is still no efficient purification method of exosomes and that the studies reporting on bioactive molecules in exosomes, despite their high significance, should be considered with caution.

In general, the topic is interesting and of high significance, yet available studies as the authors note are scarce and scattered, making such critical review timely. Furthermore, the writing was clear, engaging, and informational. There were, however, some minor issues that the authors may want to consider before publication:

  • Since the authors acknowledge the lack of an efficient exosome purification method, they should be reserved in their nomenclature and name these preparations a more generic term such as “extracellular vesicles”, as recommended by ISEV.

  • Line 34: For completion, the authors may want to include seminal plasma (https://doi.org/10.1074/mcp.RA119.001594), since this biofluid is relevant to the topic.
  • Line 197-200 are copy-pasted from ref. 48. The authors need to reformulate the sentence.

  • The section 3 is particularly compelling and I agree with the authors in their analysis (and results) that chromatography is superior method for exosome purification, but it is still far from optimal and there is still a room for optimization. The sentence line 210, “Thus, size-exclusion chromatography is an advantageous approach for the additional effective purification of exosomes from impurities.” should be complemented with a caveat “…, but there is still a room for improvement” --see https://doi.org/10.3390/ijms21155361.

  • Line 219: incomprehensible sentence: “According to most of the methods mentioned above, it should be emphasized that usually obtain samples only enriched with exosome particles, but not pure ones” could be replaced with: “According to most of the bulk purification methods mentioned above, it should be emphasized that only enriched exosome specimens can be prepared unless single-vesicle isolation approaches, which are not yet developed, are to be used.”

  • Sentence line 234 needs to be rewritten.

  • Line 247. “unfortunately, there are currently few proteomics studies on placental exosomes” replace with “unfortunately, there are currently only a few proteomics studies on placental exosomes”.

  • Paragraph numbering: Lines 322 and 357 should be 4.2 and 4.3

  • Line 136: Methods (with s) of exosome isolation.

  • Line 353: “were analyzed” instead of “were analysis”

Round 2

Reviewer 1 Report

I have detected major self-plagiarism in the last version of this manuscript. It is extremely irritating that the publisher/editorial board is still considering this manuscript to be acceptable.

As a sceintific reviewer I am no longer willing to evaluate this manuscript.